# Phosphate-Solubilizing Capacity of *Paecilomyces lilacinus* PSF7 and Optimization Using Response Surface Methodology

**DOI:** 10.3390/microorganisms11020454

**Published:** 2023-02-10

**Authors:** Xue-Li Wang, Shu-Yi Qiu, Shao-Qi Zhou, Zhi-Hu Xu, Xue-Ting Liu

**Affiliations:** 1College of Life Sciences, Guizhou University, No. 2708 Huaxi Road, 14, Guiyang 550025, China; 2College of Liquor and Food Engineering, Guizhou University, No. 2708 Huaxi Road, 14, Guiyang 550025, China; 3College of Resources and Environmental Engineering, Guizhou University, No. 2708 Huaxi Road, 14, Guiyang 550025, China

**Keywords:** *Paecilomyces lilacinus*, phosphorus-solubilizing fungi, phosphate-solubilizing capacity, medium optimization

## Abstract

Phosphorus-solubilizing microorganisms release organic acids that can chelate mineral ions or reduce the pH to solubilize insoluble phosphates for use by plants; it is important to study potential phosphorus-solubilizing microorganisms for use in agriculture. In this study, PSF7 was isolated from the soil of the Wengfu Phosphorus Tailings Dump in Fuquan City, Guizhou Province, China. PSF7 was identified as *Paecilomyces lilacinus,* based on morphological characterization and ITS sequencing analysis. The relationship between the phosphorus-solubilizing capacity and pH variation of PSF7 under liquid fermentation was studied. The results showed that there was a significant negative correlation (−0.784) between the soluble phosphorus content of PSF7 and the pH value. When PSF7 was placed under low phosphorus stress, eight organic acids were determined from fermentation broth using HPLC, of which tartaric acid and formic acid were the main organic acids. Different optimization parameters of medium components were analyzed using response surface methodology. The optimized medium components were 23.50 g/L sucrose, 1.64 g/L ammonium sulfate and soybean residue, 1.07 g/L inorganic salts, and 9.16 g/L tricalcium phosphate, with a predicted soluble phosphorus content of 123.89 mg/L. Under the optimum medium composition, the actual phosphorus-solubilizing content of PSF7 reached 122.17 mg/L. Moreover, scanning electron microscopy analysis of the sample was carried out to characterize the phosphate-solubilizing efficiency of PSF7 on mineral phosphate. The results provide useful information for the future application of PSF7 as a biological fertilizer.

## 1. Introduction

Phosphorus is an indispensable nutrient element for plant growth and development [1]. It plays an important role in plant cell division, energy transfer, signal transduction, nucleic acid synthesis, photosynthesis, and other processes [2]. Nayak et al. [3] demonstrated that the specified nutritional requirement orders in most of the age groups of plants are nitrogen > phosphorus > potassium > zinc. The phosphorus that is required by plants mainly comes from mineral or organic compounds. Compared with other nutrients, the phosphorus availability in the soil is low, ranging from 400 to 1000 mg/kg [4]. When the directly available phosphorus content is lower than 0.2% of the total phosphorus content in the soil, the plants will suffer from problems of slow development, the yellowing and blackening of leaves, and a decline in fruit rate [5]. Low phosphorus stress has similar impacts on plants as nitrogen deficiency [6].

The soluble phosphorus in the soil that can be used by plants is only 0.1% of the total phosphorus content. At present, fertilizer is mainly used to meet the phosphorus demand of crop growth [7]. However, according to the properties of the soil, fertilizers, apatite, or phosphate rock may be applied to the soil in different ways, at different times, and in different amounts. Some of the phosphorus that enters the soil is used by plants and others are combined with Ca^2+^, Mg^2+^, Fe^3+^, and Al^3+^ plasma to form insoluble phosphate in the soil [8]. Therefore, no matter whether they are applied with the single application of calcium superphosphate or the mixed use of diammonium phosphate, the fertilizer use efficiency is generally approximately 25% [9,10]. When insoluble phosphate enters the soil and accumulated in large quantities, it destroys beneficial rhizosphere microorganisms, resulting in an imbalance in the proportion of soil microorganisms, a decline in soil fertility, crop yield reductions, and other problems [11,12]. Long-term and frequent applications of phosphate fertilizer increase the risk of phosphorus leaching, accelerate algal bloom and eutrophication in rivers, and harm aquatic ecosystems [13]. In view of the many problems of chemical phosphorus fertilizer, researchers have begun to seek environmentally compatible and economically feasible alternatives to improve the growth of crops in phosphorus-deficient soils [14].

In recent years, research on phosphorus-solubilizing microorganisms (PSMs) has provided new schemes and ideas for the application of phosphorus [15,16,17]. PSMs can transform insoluble phosphate in soil compounds into soluble phosphorus [18]. At present, the reported PSMs mainly include phosphorus-solubilizing bacteria, represented by *Bacillus* and *Pseudomonas* [19,20], a small number of reports indicated that *Streptomyces* and *Micromonospora* also have the phosphorus-solubilizing capacity [21], and phosphorus-solubilizing fungi, represented by *Aspergillus*, *Penicillium*, and *Trichoderma* [22,23,24]. In addition to increasing the available phosphorus content, PSMs can synthesize iron carriers, indoleacetic acid (IAA), gibberellin (GA), cytokinin (CK), and other plant hormones and release antibacterial compounds [25,26]. Therefore, PSMs have broad application prospects in promoting plant growth, increasing crop yield, and reducing various plant diseases [27,28]. However, there are differences in the species of PSMs that are isolated from different habitats, and their phosphorus solubilization activities [29]. Therefore, the identification and evaluation of the phosphate-solubilizing capacities of PSMs are of great significance for further research.

At present, it is generally accepted that the main mechanism of phosphate solubilization by PSMs is the production of low-molecular-weight organic acids [30,31]. The common organic acids are acetic acid, citric acid, succinic acid, propionic acid, glycolic acid, malonic acid, fumaric acid, tartaric acid, glucuronic acid, and α-ketogluconic acid [32,33,34,35]; among them, gluconic acid and α-ketogluconic acid have made significant contributions in phosphate solubilization [36,37,38]. The role of such organic acids may be due to the reduction in pH, the formation of a complex with iron or aluminum in iron phosphate and aluminum phosphate, or competition with phosphate for adsorption sites in soil, thus releasing plant-available phosphate into the soil [39]. In addition, organic acids can also promote plants to obtain root nutrients, mineral weathering, microbial chemotaxis, and metal detoxification [40]. However, acidification does not seem to be the only mechanism of phosphate solubilization, as studies have confirmed that, in some cases, the increase in soluble phosphate content in the fermentation broth has no significant correlation with the decrease in pH [41]. According to reports, PSMs can also dissolve insoluble phosphorus by releasing proton/bicarbonate, cation chelation, adsorption, or the secretion of phosphatase, inorganic acid, iron carrier, phenolic compounds, and humus [42,43].

The phosphorus-solubilizing process is a complex phenomenon, which mainly depends on many other factors such as the nutritional richness of soils and the growth dynamics and physiological functions of the PSMs [44]. Moreover, the performance of PSMs has also been found to be severely affected by the composition and proportion of the medium culture [45], such as the types of carbon sources and their concentrations [46], the types of nitrogen sources [47], etc. Recently, studies have focused on isolating and screening PSMs from different rhizosphere soils and then they have evaluated their phosphate-solubilizing capacities in vitro [48,49]. However, there are few reports regarding the impacts of medium culture on the development of growth and the phosphate-solubilizing activities of PSMs. The response surface method (RSM) and statistical methods have been increasingly used to predict mathematical models to evaluate the responses of dependent variables to selected factors (independent and predictive variables). RSM is widely used in modeling processes that are affected by multiple independent variables, with the goal of optimizing the process [50].

This study aimed to isolate and to screen PSF7, which has the potential to solubilize insoluble mineral phosphate, from the soil near phosphorus tailings. The PSF7 screened in this study was identified as *Paecilomyces lilacinus. Paecilomyces* is a common saprophytic filamentous fungus, with antibacterial, antiviral, weeding, insecticidal, and cytotoxic activities [51]. However, a number of species in this genus are plant pathogens. When the classical identification methods are compared with genetic marker analysis, the identification of some entomopathogenic or nematophilous species in the genus Paecilomyces may be questioned [52]. *P. lilacinus* belongs to nematophagous fungi, which have an effective mode of biological control against plant-parasitic nematodes such as root-knot nematode eggs and cyst nematodes [53]. *P. lilacinus* strain 251 was isolated from the soil of the Philippines and it was developed as a biocontrol agent by Australian Technology Innovation Corporation Limited (ATIC). It has been proven that *P. lilacinus* strain 251 has a good degree of efficacy in the nematode control of tomato, tobacco, potato, and other plants; the study found that it did not persist long in the soil and that it would not prey on healthy plant roots [54,55,56]. Constantin et al. [57] isolated *P. lilacinus* 112 from the soil and tested the highest antifungal activity against a Cladosporium isolate, Rhizoctonia. solani, and Sclerotinia sclerotiorum. More importantly, *P. lilacinus* 112 was proven to have the ability to solubilize phosphorus and it was found that the strain 112 could promote the growth of tomato seedlings in a greenhouse experiment.

Recently, *P. lilacinus* has been often used in the field of biological control [58], with less research on its application in phosphate solubilization. Therefore, the research on the phosphate-solubilizing capacity of *P. lilacinus* should be carried out so that it can be used in the preparation of biological fertilizers in the future, not only to play its biological protection role, but also to improve the content of soluble phosphorus in soil and to promote plant growth. This research-based study aimed to elaborate on the effect of the optimization of parameters such as the type and concentration of nitrogen sources, the type and concentration of carbon sources, phosphorus source concentration, and inorganic salt concentration. The interaction between different medium components is also studied, as well as the influence on the phosphate-solubilizing capacity of strain PSF7. The functional relationships between the different variables were established using RSM and the phosphorus solubilization effect was characterized using scanning electron microscopy (SEM).

## 2. Materials and Methods

### 2.1. Characterization and Identification of the Phosphate-Solubilizing Fungi

PSF7 was isolated and screened from the soil (106°32′ E, 32°39′ N) in the Wengfu Phosphorus Tailings Dump, Fuquan City, Qiannan Buyei and Miao Autonomous Prefecture, Guizhou Province, China. The strain was preserved in the China Center for Type Culture Collection (CCTCC; Accession No. CCTCCM 2018239) and was grown on Potato Dextrose Agar (PDA) medium (potato 200 g/L, glucose 20 g/L, peptone 5 g/L, potassium dihydrogen phosphate 3 g/L, magnesium sulfate 1.5 g/L, and agar 20 g/L).

The characteristics of PSF7 were determined using the methods described in Zeng [59]: the PSF7 was inoculated on the PDA medium and cultured in a 30 °C incubator for 5 days to observe the morphological characteristics of the colony. The identification of the phosphate-solubilizing fungi is primarily based on microscopic morphology, including conidial morphology, the arrangement of conidia, the type of conidiogenous cell, and the presence of sporodochia [52]. The strain was dyed using a lactic acid stone carbonate cotton blue dye method and the morphologies of conidia and conidiophores were observed using an Olympus upright fluorescent microscope (Olympus, Tokyo, Japan) under 400× and 1000×, respectively.

The activated strain PSF7 was put into 100 mL of Potato Dextrose Broth (PDB) medium (potato extract 4 g/L, glucose 20 g/L, and pH 5.6±0.2) and the table was shaken at 30 °C, 180 r/min until mycelium balls were formed. An extract of five milliliters of spore suspension was collected and centrifuged at 12,000 r/min for 3 min under a low temperature; the supernatant was discarded, the mycelium balls were collected, and they were ground into powder after rapid freezing using liquid nitrogen. The DNA of PSF7 was extracted with reference to the fungal genomic DNA extraction kit (BIOMIGA, San Diego, CA, USA) and the fungi were PCR amplified using the universal upstream primer ITS1 (5′-TCCGTAGGAACCGCGG-3′) and downstream primer ITS4 (5′-TCCTCGCTTATTGATGCC-3′) in the fungal ITS region [60]. The PCR amplification reaction system of the fungi was 25 μL: Fungal DNA Template 2 μL, and 2 × Taq PCR Master Mix 12.5 μL. The upstream and downstream primers were 1 μL and ddH_2_O was 8.5 μL. The PCR reaction procedure was as follows: 95 °C for 2 min, followed by 33 cycles of 95 °C for 30 s, 55 °C for 30 s, and 72 °C for 2 min; extension at 72 °C for 5 min and storage at 4 °C. The amplified PCR product was prepared using 1% agarose gel × TAE electrophoresis buffer, with a loading capacity of 5 μL. The PCR products were purified using the e.Z.N.A.TM ^®^ Cycle Pure Kit (Omega, Inc., Norcross, GA, USA) and sequenced at Yinghuai Jieji Trading Co., Ltd. (Shanghai, China). The sequences were input into the National Center for Biotechnology Information (NCBI) (http://www.ncbi.nlm.nih.gov, accessed on 10 December 2022) database of the United States National Center for Biotechnology Information and were compared with the sequences in the Genbank database using the BLAST algorithm. Clustal_X was used to accomplish the sequence alignment and Mega 6.0 software was used to construct a neighbor-joining-based phylogenetic tree [61].

### 2.2. Evaluation of Phosphate-Solubilizing Capacity

To evaluate the phosphate-solubilizing capacity of PSF7, the inorganic phosphorus selective (IPS) medium (glucose 10.0 g/L, Ca_3_ (PO_4_) _2_ 5.0 g/L, (NH_4_)_2_ SO_4_ 0.5 g/L, NaCl 0.3 g/L, KCl 0.3 g/L, MgSO_4_ 0.3 g/L, FeSO_4_ 0.03 g/L, and MnSO_4_ 0.03 g/L), either broth or solid (solidified with 1.8% agar), was used and tricalcium phosphate (TCP) was the sole phosphate sources. The purified strain PSF7 was incubated in the IPS solid medium and was cultured at 30 °C for 5 days; the production of the phosphorus-solubilizing ring was observed. The diameter of the phosphorus-solubilizing ring (*D*) and the diameter of the colony (*d*) were measured using the cross method and the phosphorus-solubilizing index (PSI) [62,63].
(1)$$\mathrm{PSI}=\frac{D}{d}$$
where PSI is the solubilizing-phosphorus index, *D* is the phosphorus-solubilizing ring diameter (mm), and *d* is the colony diameter (mm).

Strain PSF7 was incubated in PDA medium for 3 days at 30 °C and the spore suspension was subsequently obtained by filtering with four layers of sterile gauze. The spore amount was approximately 1 × 10^8^ CFU/mL. Then, 1.5% of the spore suspension was inoculated into the IPS broth medium and incubated under 30 °C, 180 r/min for 7 days. At the same time, the same medium without PSF7 was used as a control. The culture supernatant was collected every day to evaluate the pH and the soluble phosphorus content [61]. The pH of the supernatants was directly measured using a pH meter (VSTAR10, Thermo Fisher, Waltham, MA, USA). The other part of the supernatants was centrifuged at 10,000× *g* for 5 min and filtered through a 0.45 μm membrane. The soluble phosphate content of the filtrates was measured using the Molybdenum Antimony Colorimetry method [64]. The OD700 of filtrates were measured using a Multiskan Spectrum Microplate Spectrophotometer (Multiskan SkyHigh, Thermo Fisher, Waltham, MA, USA) and the content of available phosphorus in the supernatant was calculated according to the standard curve of the soluble phosphorus content (y = 0.5004x − 0.0073 and correlation coefficient R^2^ = 0.9999).

### 2.3. Analysis of Organic Acids

The spore suspension of PSF7 was prepared according to the method described in Section 2.2 and 1.5 mL spore suspension was added to every 100 mL of IPS broth medium, cultured in a 180-rpm shaker at 30 °C for 5 days. The supernatant was centrifuged and filtered with 0.45 membrane for later use. Oxalic acid, tartaric acid, formic acid, malic acid, lactic acid, acetic acid, fumaric acid, citric acid, and succinic acid in the supernatants were analyzed using high-performance liquid chromatography (Agilent 1200 Liquid Chromatograph, Santa Clara, CA, USA) [65]. The chromatographic separation conditions were as follows: the chromatographic column was SHIMADZU C18-AQ (5 μm, 250 mm × 4.6 mm) and the mobile phase A was 0.02 mol/L KH_2_PO_4_, pH = 2.9; the mobile phase B was methanol with a flow rate of 0.4 mL/min and an injection volume of 100 μL, a column temperature of 32 °C, and an ultraviolet detection wavelength of 210 nm. The elution procedure for the determination of organic acids was as follows: 0 min, A phase: 100%; 8 min, A phase: 98%, B phase: 2%; 17 min, A phase: 95%, B phase: 5%; and 26 min, A phase: 100%. The atlas of the supernatant sample was compared with that of the standard sample of the organic acids mentioned above. The types and contents of the organic acids in the samples were determined according to the peak time and peak area of the standard sample.

### 2.4. Effects of Medium Composition on Phosphate-Solubilizing Capacity

The effects of different types of carbon sources (soluble starch, sucrose, glucose, maltose, fructose, and lactose) and carbon source concentrations (10, 15, 20, 25, and 30 g/L), different types of nitrogen sources (ammonium sulfate, ammonium nitrate, sodium nitrate, soybean residue, yeast extract, and ammonium chloride) and nitrogen source concentrations (0.50, 1.00, 1.50, 2.00, and 2.50 g/L), and different phosphorus source concentrations (2.5, 5.0, 7.5, 10.0, and 12.5 g/L) and different inorganic salt concentrations (0.50, 1.00, 1.50, 2.00, and 2.50 g/L) on the phosphate-solubilizing capacity of strain PSF7 were tested with experiments of a single factor using a completely randomized design. The concentration of the spore solution was set to 1 × 10^8^ CFU/mL and was inoculated into 50 mL sterilized IPS broth medium at a ratio of 1.5%. The loading capacity of each flask was 20% and it was placed in a shaker at 30 °C and 180 r/min; the culture medium without PSF7 inoculation served as a control. The experimental group and control group were repeated three times. After 5 days, 10 milliliters of fermentation supernatant was collected and centrifuged at 10,000× *g* for 5 min, then passed through a 0.45-μm filter. The content of the soluble phosphorus was determined using the method described in Section 2.2.

### 2.5. Design of the Experiment

The Box–Benhnken model of RSM in the Design-expert 12 Software (Stat-Ease, Inc., Minneapolis, MN, USA) was used to optimize the parameters, such as the types and concentration of nitrogen sources, the type and concentration of carbon sources, phosphorus sources concentration, and inorganic salt concentration. The significance and interrelationships between the sucrose concentration present in the range from 20 to 30 g/L, the concentration of ammonium sulfate and soybean residue in the range from 1 to 2 g/L, the concentration of tricalcium phosphate in the range from 7.5 to 12.5 g/L, the concentration of inorganic salt in the range from 0.5 to 1.5 g/L, and the maximum phosphate solubilization content of PSF7 under the above conditions were studied. The design and results of the Box–Benhnken experiments were presented in Table 1.

To assess the impacts of interactions between different factors in the table, the F-test was used to obtain the validity of the model and the significance of fitting was determined through the determination of the R^2^ coefficient. Finally, the composition parameters of the medium were statistically optimized using the statistical analysis of variance (ANOVA) technique. For the statistical significance of the experiment, a *p*-value as <0.05 was selected as the criterion [66].

### 2.6. Scanning Electron Microscopy

In order to explore the phosphorus-solubilizing capacity of PSF7 in practical application, phosphorus tailings containing insoluble phosphorus were selected to replace tricalcium phosphate in the IPS broth medium, and other components remained unchanged. The PSF7 spore suspension were added to the medium containing phosphorus tailings (inoculation amount: 1.5%), and cultured at 30 °C, 180 r/min for 5 days. The shaking table medium without PSF7, and the static medium without PSF7 were used as the control group. An H-800 transmission electron microscope (HITACHI) was used to observe the morphology of the phosphorus tailings powder after 5 days of culture.

### 2.7. Data Analysis

Partial data were expressed as an average of three replicates, and the results were presented as the average ± standard deviation. Significant differences were determined using ANOVA and Duncan’s test, using IBM SPSS Statistics version 25 (IBM, Armonk, NY, USA), and a *p* < 0.05 was considered to be statistically significant. Graphpad Prism 9.3 was used to draw the significant difference histogram, and MEGA6.0 was used to draw the phylogenetic tree of PSF7, using the neighboring method from the distance matrices.

## 3. Results

### 3.1. Morphological Characterization

PSF7 was cultured on PDA medium at 30 °C for 5 days; the micromorphology was observed using a microscope. The colony on PDA grew moderately slowly and the diameter reached 45–55 mm after 5 days of incubation. *Paecilomyces* were identified by having verticillate conidiophores bearing phialides [52]. The colony was regular and round and it consisted of a basal felt with aerial mycelium. At first, the front of the mycelium was white, but when sporulating, the center of the colony changed to a lilac color. The back of the colony was yellow (Figure 1a,b).

Colonies formed well-developed, colorless spores with septum branches and the septum of the central pore could promote the transport of substances. The phialides were characteristically swollen at the base and gradually narrowed into a long beak; finally, the spores were produced at the top of the phialides in shaped chains. Sporangia were subglobose and smooth and they measured 2–8 × 2–4 µm in size. Zygospores were not observed (Figure 1c,d).

### 3.2. Identification via 16S rRNA Sequence Analysis

The target fragment size of PSF7 was 589 bp (Figure 2a). The gene sequence of the sequenced PSF7 was input into the NCBI BLAST database and compared with the gene sequence of a similar, published strain. Then, MEGA6.0 software was used to analyze the phylogenetic relationship between PSF7 and close members of strains (Figure 2b). It can be seen that PSF7 and *P. lilacinus* (GQ229080.1) were clustered in the same branch and their sequence homology was 99%. Combined with the morphological characteristics of the colony, the observation of the structures of the spores, and the results of the ITS sequence analysis, PSF7 was identified as *P. lilacinus*.

### 3.3. Analysis of Phosphate-Solubilizing Capacity

The phosphate-solubilizing capacity of PSF7 was evaluated using its PSI and PSE (Table 2). The phosphorus-solubilizing halo on the culture medium is one of the criteria that determine the phosphorus-solubilizing capacity of a strain [44]. The phosphate-solubilizing capacity and the pH value of the culture medium of PSF7 changed significantly during 7 days of liquid culture (Figure 3). In the first 48 h, the pH value of the culture medium of PSF7 decreased significantly from 7.00 to 4.38. Thereafter, the pH value in the culture medium rose gradually and reached 5.10 on the seventh day. However, the content of soluble phosphorus in the culture medium of PSF7 strain first increased and then decreased gradually during 7 days of liquid culture. The content of soluble phosphorus changed significantly from 0 to 72 h and, when cultured to 72 h, the content of soluble phosphorus reached 98.09 μg/mL. After 72 h, the content of soluble phosphorus tended to be stable.

### 3.4. Analysis of Organic Acids

Organic acid secretion plays an important role in PSMs’ mineral phosphate solubilization. The types of organic acids produced by different phosphorus-solubilizing fungi are different. In this research, eight kinds of organic acids were detected in the fermentation broth of PSF7 (Table 3). Among them, tartaric acid had the highest content at 243.08 mg/L, accounting for 23.39% of the total organic acid, followed by formic acid (239.88 mg/L, 23.03%). Acetic acid had the lowest content, at 28.63 mg/L.

### 3.5. Effects of Medium Composition on Phosphate-Solubilizing Capacity

The effects of the medium components on the phosphate-solubilizing capacity of PSF7 were shown in Figure 4a–f. The types of carbon sources had different effects on the phosphate-solubilizing capacity of PSF7 (Figure 4a). When using different carbon sources, the soluble phosphorus content was in the order of sucrose > glucose > maltose > soluble starch > fructose > galactose, from high to low. Therefore, PSF7 showed a better phosphate-solubilizing capacity when sucrose was used as the carbon sources, with the soluble phosphorus content reaching 100.24 mg/L. Different nitrogen sources had a great impact on the phosphate-solubilizing capacity effects of PSF7 (Figure 4b). When using different nitrogen sources, the soluble phosphorus content was in the order of soybean residue > ammonium sulfate > ammonium nitrate > sodium nitrate > yeast extract > potassium nitrate, from high to low. When PSF7 used soybean residue and ammonium sulfate as the nitrogen sources, it showed the best phosphate-solubilizing performance, with the soluble phosphorus content reaching 114.76 mg/L and 102.36 mg/L, respectively. Based on the above results, it was found that PSF7 can use both inorganic nitrogen sources (ammonium sulfate) and organic nitrogen sources (soybean residue). Therefore, soybean residue and ammonium sulfate were selected as the nitrogen sources in this experiment and the mass ratio of soybean residue to ammonium sulfate was 1:1.

As the carbon sources can affect the synthesis of phosphatase and thus limit the phosphate-solubilizing capacity of PSMs [29], when the sucrose concentration reached 25 g/L, the soluble phosphorus content of PSF7 reached the maximum at 109.14 mg/L (Figure 4c). With the increase in sucrose concentration, the content of soluble phosphorus increased, mainly because sucrose provided carbon sources and energy for the growth of the strain. With the increase in the nitrogen source concentration, the content of soluble phosphorus increased first and then decreased. When the nitrogen sources concentration reached 1.5 g/L, the soluble phosphorus content peaked at 102.61 mg/L (Figure 4d). With the increase in the concentration of tricalcium phosphate, the soluble phosphorus content increased first and then decreased. When the concentration of tricalcium phosphate reached 10.0 g/L, the soluble phosphorus content reached the highest at 113.97 mg/L (Figure 4e). With the increase in the inorganic salt concentration, the soluble phosphorus content first increased and then decreased. When the concentration of inorganic salt reached 1.00 g/L, the soluble phosphorus content was the highest at 113.55 mg/L (Figure 4f).

### 3.6. Response Surface Test Analysis

According to Box–Behnken’s central combination test design principle, the quadratic multiple regression equation is obtained as follows:

R = 116.86 − 11.77A + 14.00B − 16.00C + 8.15D + 0.65AB + 2.92AC + 10.33AD − 5.13BC − 8.12BD − 1.51CD − 22.15A^2^ − 23.07B^2^ − 26.16C^2^ − 11.85D^2^

R is the amount of phosphate and A, B, C, and D correspond to sucrose, (ammonium sulfate and soybean residue), tricalcium phosphate, and inorganic salt, respectively.

The ANOVA result of the regression equation fitting using the response surface of phosphate solubilizing (Table 4) showed that the model F-test result was 25.42 (*p* < 0.0001). The misfit term *p* = 0.3668 > 0.05, so the model was significant. The R^2^ of the model was 0.9621, indicating that the degree of fitting was very good and the Adj R^2^ of 0.9243 showed that it had a good reliability. Therefore, this model can be used to analyze and to predict the optimal medium conditions for phosphate solubilizing.

The sucrose concentration, nitrogen concentration of ammonium sulfate soybean residue combination, tricalcium phosphate concentration, and inorganic salt concentration had significant effects on phosphate solubilization (*p* < 0.01). The degree of influence on the phosphate-solubilizing capacity was the nitrogen source concentration of ammonium sulfate and soybean residue combination > tricalcium phosphate concentration > inorganic salt concentration > sucrose concentration. The interaction of sucrose and the inorganic salt concentration had a significant effect on the soluble phosphorus content (*p* < 0.01), while the interaction of the nitrogen source combination of ammonium sulfate and soybean residue and the inorganic salt concentration had a significant effect on the soluble phosphorus content (*p* < 0.05). The interaction between the concentration of sucrose and the combined concentration of ammonium sulfate and soybean residue nitrogen source had less of a significant effect on the soluble phosphorus content (*p* > 0.05), while the interaction between the concentration of sucrose and the concentration of tricalcium phosphate had no significant effect on the soluble phosphorus content (*p* > 0.05); the interaction between the combined concentration of the ammonium sulfate and soybean residue nitrogen source and the concentration of tricalcium phosphate had no significant effect on the soluble phosphorus content (*p* > 0.05). The cross action of tricalcium phosphate concentration and inorganic salt concentration had no significant effect on the soluble phosphorus content (*p* > 0.05), which was extremely significant for the quadratic terms of the model.

The interactions between different medium components or parameters for PSF7 liquid fermentation are shown in Figure 5. The contour map of the interaction of the nitrogen source combination concentration and the tricalcium phosphate concentration of ammonium sulfate and soybean residue on the soluble phosphorus content was a flat ellipse, indicating that the interaction had a certain effect on the phosphate solubilization but this effect was not significant (Figure 5i,j). The contour plots of the interaction of the nitrogen source combination of sucrose, ammonium sulfate and soybean residue, and sucrose and also tricalcium phosphate, tricalcium phosphate, and inorganic salts on the soluble phosphorus content were all close to round, indicating that the effects on the results were not significant (Figure 5a–h,k,l). Although the effects were not significant, when one variable was fixed, the soluble phosphorus content first increased and then decreased with the increase in the other variable, although the change was not significant. The contour lines under each interaction were ellipses and approximate ellipses, indicating that the response value R was high. According to the design expert software, the optimal fermentation medium composition of PSF7 was 23.50 g/L sucrose, 1.64 g/L ammonium sulfate and soybean residue, 1.07 g/L inorganic salt, and 9.16 g/L tricalcium phosphate. Under this condition, the model predicted that the soluble phosphorus content would be 123.89 mg/L. Under the optimum medium composition, the actual phosphorus-solubilizing content of PSF7 reached 122.17 mg/L, which was close to the predicted value, showing that the model had a good reliability and could be used in subsequent experiments.

### 3.7. Scanning Electron Microscope Analysis

Under static culture, the surface of sterile phosphorus tailings without PSF7 was obviously angular, showing a typical mineral state (Figure 6a). The surface of sterile phosphorus tailings without PSF7 was smaller in the shaking table culture than in the static culture state, with the characteristics of clastic particles (Figure 6b). After PSF7 was introduced, the strain grew rapidly under a shaking state and the mycelium formed gathered small particles of phosphorus tailings into large particles, leading to an increase in the particle shape of the phosphorus tailings, with an obvious honeycomb shape on the surface (Figure 6c). The phosphorus tailings without PSF7 did not appear to have honeycomb-like characteristics and the centrifugal force of the shaking table alone could only cause the phosphorus tailing particles to become smaller and thinner, but it maintained the typical mineral characteristics. It could be seen that PSF7 has a strong degree of solubility on insoluble phosphate rock powder.

## 4. Discussion

Phosphorus is an essential nutrient and an important limiting factor for plants [67]. In the early days of the 1990s, many researchers isolated varieties of PSMs from different samples. According to the morphological characteristics and ITS sequencing, PSF7 was identified as *P. lilacinus*. The *Paecilomyces* genus has been reported to have the similar functions. Cavello et al. [68] have found *Paecilomyces marquandii* with an efficient phosphate-solubilizing capacity, which can be grown in media with a wide range of carbon and nitrogen sources.

The original method for analyzing phosphate-solubilizing capacities was the formation of a halo around the PSMS colonies; there was always a direct correlation between the halo size and the quantitative nature of the phosphate-solubilizing capacity [69]. Constantin et al. [57] demonstrated the phosphate-solubilizing capacity of *P. lilacinus* 112 via the formation of a transparent zone surrounding the fungal colonies. Li et al. [70] found that a PSI parameter of more than or equal to 1.5 indicated a strong phosphate-solubilizing capacity, while a PSI of between 1.0 and 1.5 indicated a weak capacity. The PSI parameter of PSF7 was 1.15, according to the above research, PSF7 had an average phosphorus-solubilizing capacity. However, in recent years, much research has found that, when tricalcium phosphate is used as the sole phosphorus source, PSMs form an obvious phosphorus-solubilizing halo on the solid medium, but, when iron phosphate or aluminum phosphate are added into the medium, the PSMs have no phosphorus-solubilizing halo on the solid medium, though certain soluble phosphorus contents are detected in the liquid medium [71,72].

Therefore, more and more researchers select the content of soluble phosphorus in the fermentation broth as the standard for phosphorus-solubility capacity. Qureshi et al. [73] isolated 16 strains of PSMs from subtropical soil, which were identified as *Aspergillus* spp., *Penicillium* spp., and *Talaromyces* spp. The highest of soluble phosphorus content for each genus was 759.5 mg/L, 397.2 mg/L, and 308.0 mg/L. Wang et al. [74] isolated a *Penicillium oxalicum* strain Y2 from the soil. With Ca_3_ (PO_4_) _2_ as the sole phosphate source, Y2 can release up to 2090 mg/L of soluble phosphate within 12 days of culture. Doilom et al. [75] isolated an *Aspergillus hydei* sp. KUMCC 18-0196 from the air and determined that the maximum content of the soluble phosphorus was 1523.33 ± 47.87 mg/L. In conclusion, the PSMs screened at present were mainly *Penicillium* and *Aspergillus.* Hernandez et al. reported for the first time the phosphate-solubilizing ability of *P. lilacinus* for iron phosphate and tricalcium phosphate. The research results showed that, when the strain used iron phosphate as the substrate, the maximum content of the soluble phosphorus was 1.75 mg/L. In contrast, when tricalcium phosphate was used as the substrate, the maximum content of the soluble phosphorus was 71.28 mg/L. When PSF7 screened in this study used tricalcium phosphate as the substrate, the maximum content of the soluble phosphorus was 122.17 mg/L, showing a significantly stronger phosphate-solubilizing capacity than in Hernandez et al. [76].

In the dynamic monitoring of the content of soluble phosphorus and the change of pH, it was found that the pH of the fermentation broth decreased rapidly from 0 to 48 h and it reached the lowest value at 48 h. This may be due to the production of various organic acids, such as malic acid and citric acid, during the growth and metabolism of the phosphate-solubilizing fungi, which led to a rapid decrease in the pH of the culture solution. However, the soluble phosphorus content rose rapidly from 24–48 h, with a rise slope of 2.9. After 48 h, the slope of the increase curve of the soluble phosphorus content decreased to 0.93. The phosphate-solubilizing capacity peaked at 72 h and then the soluble phosphorus content decreased continually. Correlation analysis revealed a significant negative correlation (−0.784) between pH and phosphate solubilization. Past studies also showed that the change in pH had a negative correlation with the soluble phosphorus content [77]. Mehta [78] regressed the data and found that the maximum correlation coefficient between the pH value and the amount of soluble phosphorus by *Bacillus subtilis* CKAM was −0.96. Aikaterini et al. [79] showed that the acidification of the culture solution of *Penicillium* in the phosphate solubilization may be caused by the release of protons to balance the anions in free organic acids during the culture process.

At present, most researchers believe that the mechanism of phosphate solubilization was secreted to the low-molecular-weight organic acids. Wei et al. [80] detected oxalic acid, lactic acid, succinic acid, and acetic acid from a PSM culture medium when exploring phosphate solubilization. In addition, Illmer et al. [81] found that *Pseudomonas* sp. can secrete gluconic acid, oxalic acid, citric acid, tartaric acid, lactic acid, succinic acid, and acetic acid. Wang et al. [74] found that *P. oxalicum* secreted a large amount of oxalic acid, citric acid, and malic acid during phosphate solubilization. Nelofer et al. [82] reported that *Aspergillus niger* detected oxalic acid, succinic acid, ascorbic acid, and other organic acids in the liquid phosphate-solubilizing medium. This study has found that PSF7 secreted eight kinds of organic acids in the phosphate-solubilizing process and the contents from high to low were in the order of tartaric acid, formic acid, malic acid, oxalic acid, citric acid, lactic acid, fumaric acid, and acetic acid. The types of organic acids were roughly the same as those in previous studies. In fact, different kinds of organic acids have different phosphorus-solubilizing mechanisms. Teng et al. [83] confirmed that the concentrations of each organic acid secreted by the strain were different and that the five strains studied could secrete gluconic acid, oxalic acid, malonic acid, acetic acid, formic acid, citric acid, and succinic acid. Formic acid has been detected in many PSMs phosphate-solubilizing mediums, which shows that formic acid plays an important role in phosphate solubilization [84]. In this study, tartaric acid and formic acid were the main organic acids secreted by PSF7.

The phosphate-solubilizing capacity of PSMs is affected by the culture time, inoculation amount, temperature, carbon and nitrogen sources and their contents, and initial concentration of phosphorus [84,85]. This study investigated the effect of different kinds of carbon sources on the phosphate-solubilizing capacity of PSF7. The results showed that *P. lilacinus* PSF7 had a better phosphate-solubilizing capacity when using sucrose as the carbon sources, with the phosphate-solubilizing capacity reaching 100.24 mg/L. This result was contrary to that of Panda et al., who found that, in the medium containing sucrose, the phosphate-solubilizing capacity was two-fold lower than that of the carbon source glucose [86]. However, Reyes et al. [87] found that the best carbon source of *Paecilomyces rugulosum* was sucrose, which could be used to solubilize hydroxyapatite and FeSO_4_. Tahir et al. [88] investigated the metabolic process of the *Bacillus* T-34 strain using glucose, galactose, maltose, and sucrose as carbon sources and found that the concentrations of citric acid, malic acid, acetic acid, and lactic acid produced by strain T-34 were much higher than those produced by *Azospirillum* WS-1 and *Enterobacter* T-41. Others have acquired similar results, such as Chen et al. [89], to Archana et al. [90] who reported that *Azospirillum*, *Bacillus*, and *Enterobacter* produced different amounts of citric acid, oxalic acid, gluconic acid, and 2-ketogluconic acid during metabolism. In addition to the types of carbon sources, the concentration of the carbon sources will also affect the phosphate-solubilizing capacity of PSMs. This study investigated the effects of different sucrose concentrations on the content of soluble phosphorus using *P. lilacinus* PSF7. It was found that, with an increase in sucrose concentration, the content of soluble phosphorus increased first and then decreased. When the sucrose concentration was 25 g/L, the content of soluble phosphorus reached the maximum. This finding was the same as that of Song et al. [32]. Generally, the phosphate-solubilizing capacity of PSMs increased with an increasing amount of sucrose concentration added to the growth medium. When the sucrose concentration rose from 1% to 3%, the soluble phosphorus content of PSMs increased significantly, mainly because the increased carbon source concentration led to an increase in PSMs biomass, which led to a decrease in the acidity of the fermentation broth. However, when the carbon source concentration was 5%, the soluble phosphorus content decreased.

The impact of the nitrogen sources on PSMs was similar to that of the carbon sources. Most PSMs can only use ammonium nitrogen, nitrite nitrogen, nitrate nitrogen, or amino nitrogen as the nitrogen sources and the selection of the nitrogen sources is determined by the enzymes in the microbial system [44]. In this study, ammonium sulfate, potassium nitrate, sodium nitrate, soybean residue, yeast extract, and ammonium nitrate were used as nitrogen sources and they were tested using a liquid fermentation test. The results showed that PSF7 had a good phosphate-solubilizing performance when using soybean residue and ammonium sulfate as the nitrogen sources, with the soluble phosphorus content reaching 114.76 mg/L and 102.36 mg/L, respectively. Thus, soybean residue and ammonium sulfate were selected as the nitrogen sources. As an organic nitrogen sources, the soybean residue is not as easily absorbed by PSMs as ammonium sulfate, so it has the effect of a slow release and a slow absorption. In the research of inorganic nitrogen sources on the phosphate-solubilizing capacity of PSMs, ammonium was found to be a better nitrogen sources than nitrate. Sang et al. [61] found that strain L4 had the highest phosphorus-solubilizing content when ammonium sulfate was used as the nitrogen sources. The research of Sulbaran et al. [91] research found that the pH value of the fermentation broth of *P. agglomerates* MMB051 in the presence of KNO_3_ was 5.10 ± 0.15, while, when the nitrogen sources was changed to (NH_4_)_2_SO_4_, the pH value of the fermentation broth was 2.86 ± 0.21 and the content of the soluble phosphorus related to the change of pH value also changed. When (NH_4_)_2_SO_4_ was used as the nitrogen sources, the content of soluble phosphorus was 95.75 mg/L, while, when KNO_3_ was used as the nitrogen sources, the content of soluble phosphorus was 58.15 mg/L. Habte et al. [47] found that, when NH_4_Cl was used as the nitrogen sources, the content of soluble phosphorus increased with the increasing NH_4_Cl concentration. However, when the concentration of NH_4_Cl was too high, the excessive NH4^+^ content had a negative impact on the growth of *Morella* sp.

Adding inorganic salts can significantly increase the quantity of conidia in the fermentation broth [92]. In this study, NaCl, MgSO_4_·7H_2_O, FeSO_4_·7H_2_O, and K_2_HPO_4_ were used as the combination of inorganic salts. The components in the inorganic salts were configured according to the mass ratio of 1:1:1:1. The soluble phosphorus content increased with the increase in the concentration of inorganic salts up to a concentration of 1.00 g/L, the soluble phosphorus content reached the maximum of 113.55 mg/L. With the increase in the mass concentration of inorganic salts, the soluble phosphorus content increased first and then decreased, which may be related to the chelation of metal ions in the PSMs’ phosphate-solubilizing mechanism [93].

In this study, Ca_3_ (PO_4_) _2_ was used as the only phosphate source to investigate the effect of different phosphate concentrations on the phosphate-solubilizing capacity of PSF7. When the content of tricalcium phosphate was 2.5–5.0 g/L, the phosphate-solubilizing capacity of PSF7 was low. With the increase in the tricalcium phosphate concentration, the phosphorus-solubilizing capacity of PSF7 also significantly increased. When the concentration of tricalcium phosphate reached 12.5 g/L, PSF7 was inhibited, resulting in a decrease in the phosphorus-solubilizing capacity in the final fermentation broth.

In fact, except for the carbon source, nitrogen source, etc., the initial pH, initial inoculum concentration, and cultivation temperature affected the phosphorus-solubilizing capacity [94,95]. Zhang et al. [96] found that, when the initial inoculum was 4× 10^4^ CFU/mL, the phosphorus-solubilizing capacity reached the maximum. Osorno et al. [97] showed that the inoculation with *Mortierella* sp. significantly decreased (*p* < 0.05) the solution pH and increased the soluble phosphorus content. However, larger amounts of inoculum generated significantly less soluble phosphorus content. Chai et al. [98] confirmed that the biomass of the inorganic phosphorus-solubilizing bacteria was more important than the cell physiology. In this study, in order to ensure the consistency of the biomass of PSF7 in each experiment, the inoculum amount was set as 1.5%; the effect of the biomass on the phosphorus-solubilizing capacity will be the focus of future research.

In this study, SEM was used to confirm the phosphorus-solubilizing efficiency of PSF7 in the medium containing phosphorus tailings. The results found that the mineral surface of the phosphorus tailings changed significantly after inoculation with PSF7, which had an obvious honeycomb shape on the surface. Xu et al. [99] showed that raw phosphate rock powders contained large spherical particles of varying sizes (10–50 μm) with regular or rough surfaces and fragmentations. Qiu et al. [100] found that the *Aspergillus niger* was cultured for 20 days in the medium with calcium phosphate and the SEM images showed that a large number of hyphae were found in the calcium phosphate sample and that calcium phosphate was wrapped, dispersed, or attached to the hyphae. The particles existed in the form of a diamond, with a short columnar structure, and the aggregates were chrysanthemum. Therefore, PSF7 is expected to be used as a biological fertilizer to improve the content of the available phosphorus in soil and to promote plant growth.

## 5. Conclusions

PSF7 was isolated from the soil of the Wengfu Phosphorus Tailings Dump and it was identified using morphological characterization and ITS sequencing analysis. PSF7 was identified as *P. lilacinus;* most of this genus are used as biological control agents. In this study, PSF7 showed a high mineral phosphate-solubilizing capacity. In the solubilizing process, PSF7 secreted oxalate acid, tartaric acid, formic acid, malic acid, lactic acid, acetic acid, and fumaric acid; citric acid, tartaric acid, and formic acid were the main organic acids secreted by PSF7. RSM was used to optimize the medium composition of PSF7. The study found that, when the medium was sucrose at 23.50 g/L, ammonium sulfate and soybean residue at 1.64 g/L, inorganic salts at 1.07 g/L, and tricalcium phosphate at 9.16 g/L, the predicted soluble phosphorus content of PSF7 was 123.89 mg/L. Under the optimum medium composition, the actual phosphorus-solubilizing content of PSF7 reached 122.17 mg/L. The results are helpful for better understanding the phosphate-solubilizing mechanism of PSMs and for providing useful information for the future application of PSF7 as a biological fertilizer.

## Figures and Tables

**Figure 1 microorganisms-11-00454-f001:**
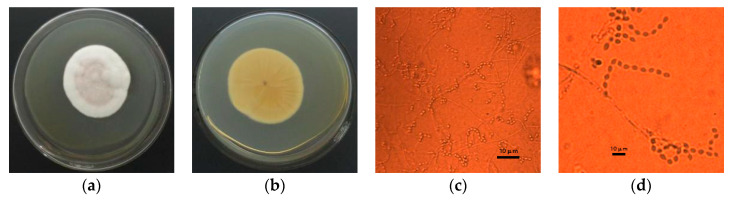
Morphological characteristics of PBF7. (**a**) Positive side of culture medium colony. (**b**) Back of the culture medium colony. (**c**) Micrograph (400×). (**d**) Micrograph (1000×).

**Figure 2 microorganisms-11-00454-f002:**
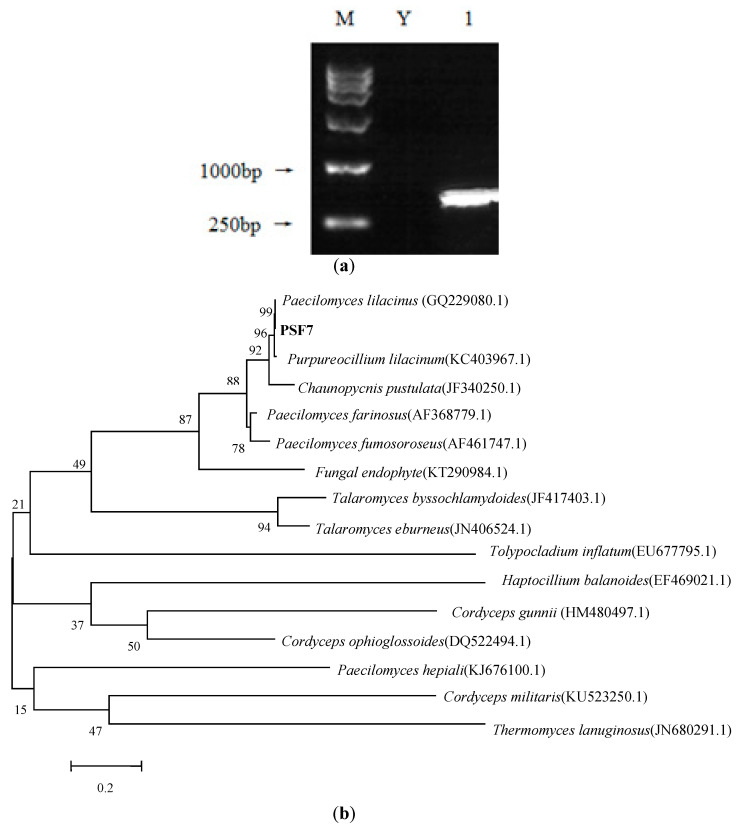
(**a**) Electrophoresis map of ITS sequence PCR amplification of PSF7 and (**b**) phylogenetic tree of PSF7 based on ITS sequences. The sequence numbers in parentheses correspond to the GenBank accession numbers of the reference strains; the numbers at branch points represent confidence level; and Bar, 0.2 substitutions per nucleotide positions.

**Figure 3 microorganisms-11-00454-f003:**
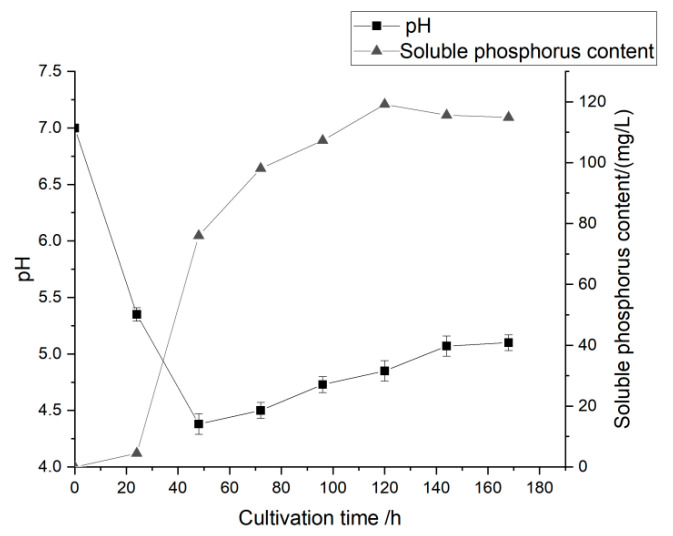
Changes of pH value and phosphate-solubilizing capacity in liquid culture of PSF7.

**Figure 4 microorganisms-11-00454-f004:**
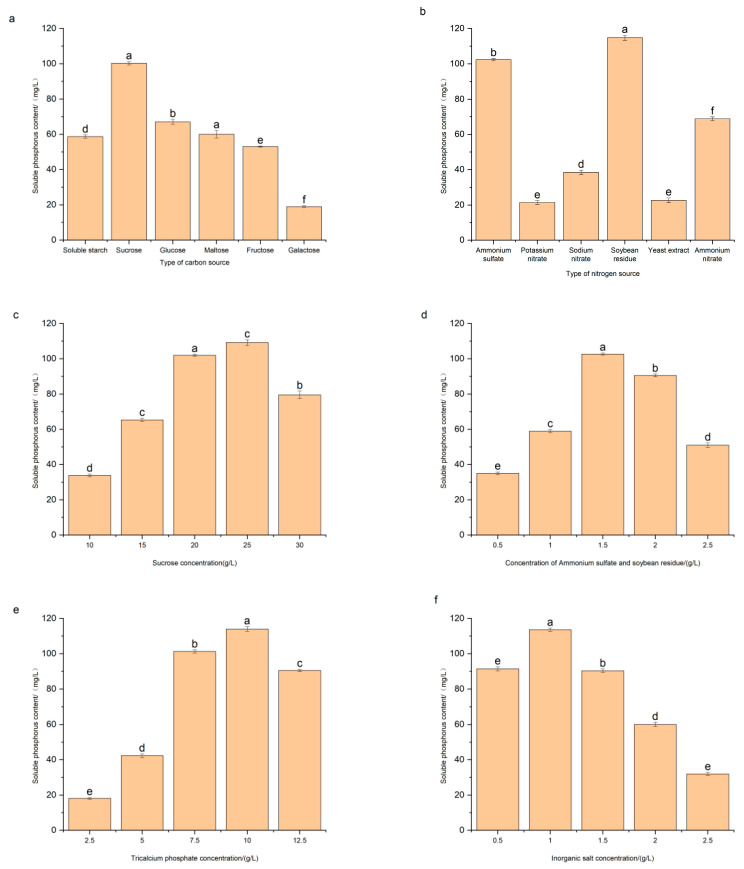
Effect of medium composition on the phosphate-solubilizing capacity of PSF7. (**a**) Types of carbon sources. (**b**) Types of nitrogen source. (**c**) Sucrose concentration. (**d**) Concentration of ammonium sulfate and soybean residue. (**e**) Tricalcium phosphate concentration. (**f**) Inorganic salt concentration.

**Figure 5 microorganisms-11-00454-f005:**
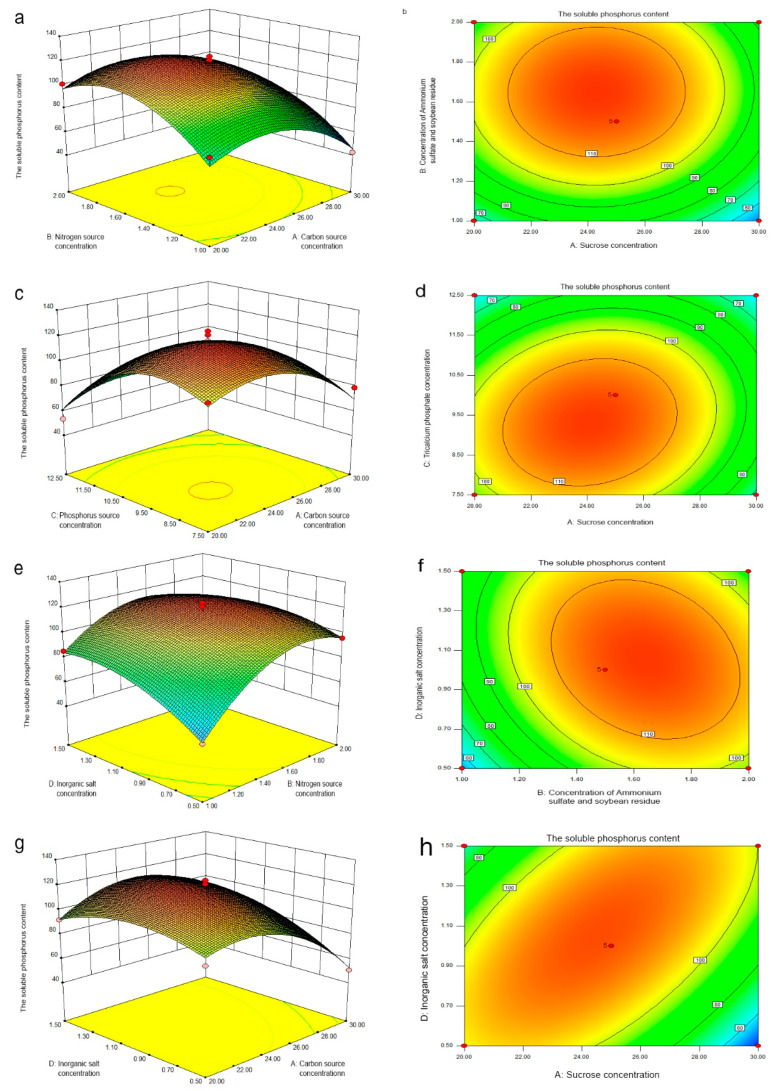
Response surface plots and contour lines of effects for the interaction between factors for the phosphate-solubilizing capacity of PSF7. (**a**) Response surface plots of factor AB. (**b**) Contour line of factor AB. (**c**) Response surface plots of factor AC. (**d**) Contour line of factor AC. (**e**) Response surface plots of factor BD. (**f**) Contour line of factor BD. (**g**) Response surface plots of factor AD. (**h**) Contour line of factor AD. (**i**) Response surface plots of factor BC. (**j**) Contour line of factor BC. (**k**) Response surface plots of factor CD. (**l**) Contour line of factor CD.

**Figure 6 microorganisms-11-00454-f006:**
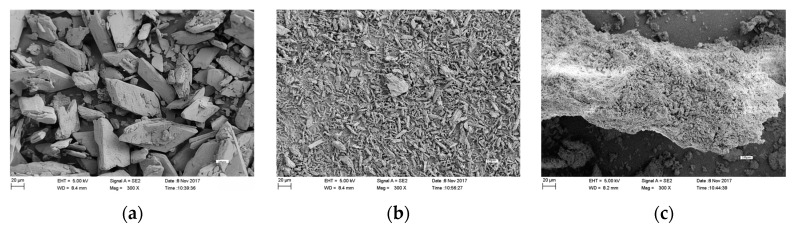
(**a**) Phosphate tailing powder in static culture without PSF7. (**b**) Phosphate tailing powder in table culture without PSF7. (**c**) Phosphate tailing powder after adding PSF7.

**Table 1 microorganisms-11-00454-t001:** Design and results of Box–Benhnken experiment.

Run	Concentration of Sucrose	Concentrations of Ammonium Sulfate and Soybean Residue	Concentration of Tricalcium Phosphate	Concentration of Inorganic Salt	Phosphate-Solubilizing Capacity (mg/L)
	0	−1	0	1	85.47
2	0	0	1	1	65.77
3	0	1	1	0	65.99
4	1	0	−1	0	79.00
5	0	1	−1	0	96.67
6	1	−1	0	0	42.09
7	0	0	−1	−1	90.00
8	0	−1	0	−1	50.95
9	0	0	0	0	120.67
10	−1	0	0	1	92.16
11	1	0	0	−1	50.44
12	1	0	0	1	93.00
13	0	−1	−1	0	56.57
14	1	1	0	0	68.76
15	−1	−1	0	0	76.91
16	0	−1	1	0	46.42
17	0	1	0	1	98.05
18	0	0	1	−1	60.09
19	−1	0	1	0	53.62
20	0	0	0	0	123.72
21	0	0	0	0	116.12
22	−1	0	0	−1	90.93
23	1	0	1	0	42.17
24	0	0	−1	1	101.72
25	0	1	0	−1	96.00
26	0	0	0	0	108.23
27	−1	1	0	0	100.98
28	−1	0	−1	0	102.13
29	0	0	0	0	115.55

**Table 2 microorganisms-11-00454-t002:** Measurement of phosphate-solubilizing capacity of the strain using the transparency circle method.

Strain	d (mm)	D (mm)	PSI
PSF7	1.99 ± 0.09 b	2.29 ± 0.10 b	1.15 ± 0.003 a

Table of abbreviations: phosphorus-solubilizing index (PSI). Notes: Significant differences are indicated by letters, same letters, no significant difference; different letters, significant difference.

**Table 3 microorganisms-11-00454-t003:** Types and concentrations of organic acids secreted by PSF7.

	Oxalate Acid	Tartaric Acid	Formic Acid	Malic Acid	Lactic Acid	Acetic Acid	Fumaric Acid	Citric Acid
Organic acid concentration (mg/L)	168.66 ± 4.07	243.08 ± 15.36	239.88 ± 8.97	201.50 ± 7.44	56.29 ± 16.17	28.63 ± 12.02	45.09 ± 18.27	58.43 ± 13.55

**Table 4 microorganisms-11-00454-t004:** Variance analysis of response surface-fitting regression equation for phosphate solubilization.

Source of Variance	Sum of Squares	Freedom	Variance	F Value	*p*	Significance
Regression model	16,817.49	14	1201.25	25.42	<0.0001	significant
A	1663.1	1	1663.1	35.19	<0.0001	**
B	2353.12	1	2353.12	49.79	<0.0001	**
C	3072.96	1	3072.96	65.02	<0.0001	**
D	796.42	1	796.42	16.85	0.0011	**
AB	1.69	1	1.69	0.036	0.8527	
AC	34.11	1	34.11	0.72	0.4099	
AD	427.04	1	427.04	9.04	0.0094	**
BC	105.37	1	105.37	2.23	0.1576	
BD	263.58	1	263.58	5.58	0.0332	*
CD	9.12	1	9.12	0.19	0.6672	
A^2^	3182.61	1	3182.61	67.34	<0.0001	**
B^2^	3451.35	1	3451.35	73.02	<0.0001	**
C^2^	4437.53	1	4437.53	93.89	<0.0001	**
D^2^	911.14	1	911.14	19.28	0.0006	**
Residual	661.68	14	47.26			
Misfit term	523.36	10	52.34	1.51	0.3668	
Pure error	138.32	4	34.58			
Total	17,479.17	28				

Note: * means significant difference (*p* < 0.05). ** Indicates that the difference is very significant (*p* < 0.01). (A) Sucrose concentration. (B) Concentration of ammonium sulfate soybean residue combination. (C) Tricalcium phosphate concentration. (D) Inorganic salt concentration.

## Data Availability

Not applicable.

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
