# Peer review of "Phosphate-Solubilizing Capacity of Paecilomyces lilacinus PSF7 and Optimization Using Response Surface Methodology"

_microorganisms, 2023, doi:10.3390/microorganisms11020454_

Round 1
Reviewer 1 Report
The reviewed article is of general interest, with a good experimental design and results that serve the community of microbiologists who want to have evidence of different communities of P-solubilizing microorganisms, one of the currently most scarce elements in mineral fertilizers. In my opinion, the part in which the strain is described in relation to morphology is very poor and leaves much to be desired for those who want to isolate this microorganism and see its characteristics. It would be highly desirable that the authors make more efforts in a better description of its morphology, microscopic photographs do not allow appreciations.
There is a difficulty in reading that, is not severe but should be improved because some sentences have the structure of a possible non-scientific translator and are difficult to read in the first part of the manuscript, they are understood but not fluently and it takes away concentration from what one is is interpreting. For example, many sentences are hyper-repetitive and can simplify be improved with codes or by deleting the repetitions that the reader finds in the Tables or Figures, it is not necessary that they recite all the treatments that we can read later. That will greatly speed up the quality of the work in its fluency. There are some flaws in the text such as careless omissions and in the scientific names, it is actually the most prevailing. They will have to pay for the services of a good style editor because otherwise they are not ready to release themselves to the magazine. There are other comments about data, which are also minor that should be addressed.
Marks in yellow are related to editing the text, either because it is incomprehensible or the sentence is badly structured, or because there are syntax spaces, periods that were omitted, misspelled scientific names, or subperindexes. Changes will be found throughout the text.

Author Response
We greatly appreciate your recognition of our work and your excellent suggestions for improving the quality of our manuscripts. We have revised the receipt to meet your requirements. Plesase see the attachment.

Reviewer 2 Report
The subject of the manuscript corresponds to the subject of the journal. The data presented in the manuscript have a significant degree of novelty and may be interest to readers. The presented results are well processed, illustrated and discussed. I would like to recommend accepting the manuscript for publication in its current form.
Author Response
We greatly appreciate your recognition of our work and your excellent suggestions for improving the quality of our manuscripts.
Reviewer 3 Report
Revision : Microorganisms / Manuscript ID: 219744
Abstract
Page1 line 12: Is it ‘flora’ or microorganisms / microbial system?
Introduction:
The importance of the paper in the introduction part can be emphasized in a better way. Authors should explain why did they focused on Paecilomyces lilacinus. A number of species in this genus belong to the plant pathogens.
Page1 line 32: ‘lower than 0.2 %’ of what
Page1 line 42: ‘it destroy rhizosphere microorganisms’ why, can Authors explain this?
Page2 line 54: Actinomycetes belong to the bacteria
Materials and methods:
Page3 line 11: Why the temperature was so high?
Results:
Page 11: Authors should indicate the Tables numbers and Figures numbers, where ever required
For tables and figures:
Figure 1 c and d, Figure 6: need the scale bar.
Figure 2, Table 2, Figures and tables should be clear and logic without the text.
Figure 2, 3, 4, 5 are unreadable (You can not read it in print)
Discussion:
Authors should indicate the Tables numbers and Figures numbers, where ever required (where ever the results are mentioned). All results needs to be discussed.
References:
The references are up-to-date but please check it carefully. The ‘References’ part needs to be prepared in accordance with the principles of editing

Author Response

(The authors gave the same response as above.)

Round 2
Reviewer 3 Report
I would like to thank the Authors for response to all received comments.
Generally I accept the manuscript in the present form.
See also Page 13 line 447: delete (Figure 5),
In figure 4 and 5 the axis title are not readible in print.